# Diagnostic and Management Difficulty of Bleeding Aorto-Duodenal Fistula Associated with Hodgkin’s Lymphoma

**DOI:** 10.3390/diagnostics11030389

**Published:** 2021-02-25

**Authors:** Boaz Nachmias, Allan I. Bloom, Alexander Gural

**Affiliations:** 1Hematology Department, Hadassah Medical Center, Jerusalem 911200, Israel; Gural@hadassah.org.il; 2Department of Medical Imaging-Interventional Radiology, Hadassah Medical Center, Jerusalem 911200, Israel; Allan@hadassah.org.il

**Keywords:** Hodgkin’s lymphoma, aortic-duodenal fistula, massive bleeding protocol

## Abstract

Primary aorto-enteric fistula (AEF) resulting from abdominal malignancy is a rare and often fatal complication. The few reports to date are mostly secondary to solid tumors. We present a case of a patient with refractory Hodgkin’s lymphoma who developed life-threatening AEF. We describe the diagnostic and therapeutic efforts, requiring a multi-disciplinary team of interventional radiology, gastroenterology, and transfusion medicine, resulting in a favorable outcome. Importantly, we offer several insights regarding the identification and management of high-risk patients, with an emphasis on pre-treatment considerations and urgent diagnosis and intervention.

## 1. Introduction

Aorto-enteric fistula (AEF) is a rare and often fatal condition associated with massive gastrointestinal (GI) bleeding. AEF can be classified as primary or secondary, where the latter implies a history of previous aortic surgery or trauma. Primary AEF is extremely rare, with a reported incidence of 0.04%, and is almost always associated with abdominal aortic aneurysm [1]. Primary malignancy-related AEF has been reported in a minority of cases, mostly as a complication of solid tumors, such as carcinoma of the pancreas and the gastrointestinal tract, or metastases from other abdominal tumors [1,2,3]. Reports of AEF as a complication of lymphoma are rare. We found a single case report describing AEF in a patient with diffuse large B cell lymphoma, which presented as a large mass encasing the aorta and parts of the duodenum [4]. Of note, the patient in that report had a known thoraco-abdominal aneurysm prior to the diagnosis of lymphoma. Two other reports described a patient with non-Hodgkin’s lymphoma involving the small bowel with suspected AEF [5], and a patient with anaplastic T cell lymphoma in the mediastinum, who develop aorto-esophageal fistula, following initiation of treatment [6].

Here, we present a report of a young patient with refractory Hodgkin’s lymphoma involving the retroperitoneum who developed massive GI bleeding from tumor-associated aorto-enteric fistula. This case highlights the complexity of diagnosis and management of AEF, and raises the question of chemotherapy protocol modification in high-risk patients.

## 2. Case Presentation

A 29-year-old male patient suffering from Hodgkin’s lymphoma, which relapsed after multiple lines of therapy, including autologous bone marrow transplantation, pembrolizumab, and brentuximab vedotin, presented with rapid intra-abdominal progression (Figure 1), and received one course of ICE (Ifosphamide, Cis-platinum, Etoposide) and one course of GDP (Gemcitabine, Dexamethasone and Cis-platinum) chemotherapy protocols. One week following the last dose of therapy, he arrived at the Hematology Day Care because of severe weakness, with hemoglobin of 5.7 gr%, but no signs of active bleeding. After receiving three units of packed red blood cells (PRBC) he was discharged home in a stable condition. The day after discharge, he presented to the emergency department with hematemesis, rectal bleeding, and anemia of 7.0 gr%. After administration of two units of PRBC, an upper endoscopic evaluation was performed, demonstrating a small superficial distal esophageal laceration considered to be the source of the hemorrhage, and managed by local adrenalin injection. Later that night, the patient developed massive hematochezia, necessitating a transfusion of an additional three units of PRBC, and was transferred to the ICU, where he remained in a stable condition for additional 24 h. A repeated upper endoscopy has demonstrated no evidence of active bleeding. However, a few hours after the procedure, he again developed massive rectal bleeding, this time with severe hemodynamic instability.

Urgent visceral angiography of celiac, superior mesenteric, left gastric, and gastroduodenal arteries was performed revealing an active source of arterial hemorrhage from a peripheral branch of the left gastric artery in the area of the gastric fundus, managed by selective coil embolization. Despite embolization, severe hematemesis and hematochezia continued, causing persistent hemodynamic instability and prompting the performance of a lateral abdominal aortography, which demonstrated active extravasation of blood into the intestine, diagnostic of an aorto-duodenal fistula (Figure 2a). Temporary balloon occlusion of the aorta was performed to stabilize the patient and control the hemorrhage. The fistula was then closed using a balloon-expandable covered stent (BeGraft, Bentley Innomed GmbH, Hechingen, Germany) at the site of the defect in the aortic wall, with immediate hemodynamic stabilization (Figure 2b). During the entire procedure, the patient received aggressive blood component support to prevent exsanguination, with 27 units of PRBC (Packed Red Blood Cells), 15 units of platelets, 22 units of FFP (Fresh Frozen Plasma) and 20 units of Cryoprecipitate administered over 3 h, in compliance with the institutional massive transfusion protocol (1PRBC:1FFP:1Platlet).

Antibiotic therapy was instituted and continued prophylactically for several weeks. The patient was discharged home in a very good general condition (ECOG1) and nine months following the procedure underwent a repeated autologous bone marrow transplantation without evidence of bleeding. The patient then received nivolumab with partial response and proceeded to a second autologous bone marrow transplant with a cyclophosphamide/total body irradiation (Cy/TBI) conditioning. The decision to proceed to a second autologous transplant was made due to inadequate conditioning with melphalan in the first transplant, the rather low success rate with allogeneic transplant in Hodgkin’s lymphoma with active disease and the general condition of the patient. Imaging following the transplant showed that there was an almost complete response at all of the sites of active disease, with residual disease in the lung. Unfortunately, a few months later he succumbed to respiratory infection, without clinical or imaging evidence of leak, aneurysm, thrombosis, or infection at the site of the covered stent.

## 3. Discussion

Primary malignancy-related AEF is a rare and mostly fatal complication, making the diagnosis and management of patients extremely challenging. Our report, as well as other reports in the literature, highlight several issues to be addressed when dealing with suspected AEF: identification of high-risk patients, attention to previous chemotherapy regimens and treatment modifications, careful review of all imaging studies, the choice of diagnostic procedures in case of suspected fistula, and an emergency management of apparent bleeding.

Certain disease and patient characteristics carry a high risk for the development of AEF. Among these are an encasement by tumor of the aorta and the GI tract, especially at the level of the duodenum due to anatomical proximity, older age, and pre-existing aortic disease, such as aortic aneurysm. Identification of high-risk patients at the time of diagnosis might necessitate preventive intervention or modification of the treatment protocol. For example, if possible, a repair of pre-existing abdominal aneurysm should be considered prior to chemotherapy. Lymphomas in particular respond rapidly to the initiation of therapy, resulting in tissue gaps with increased risk of sepsis or bleeding. Of note, similar to our case, in previously reported cases of lymphoma-related AEF, bleeding was associated with treatment initiation [4,6]. Thus, we suggest a pre-phase of steroids or reducing the dose of chemotherapy in the first cycle, to allow for better tissue repair, and to reduce the chance of neutropenia or thrombocytopenia, in case of an emergent abdominal intervention. We also advise for the high-risk patients to be admitted during the first days of therapy, for close monitoring and rapid imaging studies as necessary.

Diagnosis of AEF is often challenging and requires a high index of suspicion. Absence of the classic clinical triad of abdominal pain, hemorrhage and pulsatile mass should not diminish a high index of clinical suspicion for the development of AEF, because it is often only partially present or entirely missing [1,7]. GI bleeding is the most common clinical presentation and should prompt immediate imaging studies and close monitoring, even if initially not significant [8], where mainstay diagnostic modalities include upper endoscopy, contrast-enhanced CT scan, and catheter angiography, which also offers a therapeutic option.

Once active bleeding from an AEF is diagnosed, therapeutic efforts should be directed at hemostasis, correction of thrombocytopenia and coagulopathies (either the result of previous chemotherapy or developing in the context of massive transfusion [9]), administration of antibiotics and, eventually, aortic repair.

The existence of institutional massive transfusion protocol and its effective implementation by the blood bank team under the supervision of an experienced transfusion specialist is a necessary pre-requisite for a successful management of any life-threatening bleeding [10], with aorto-enteric fistula being possibly the most dramatic case in a non-trauma setting [11].

Several small series compared open-repair of AEF to an endovascular (EV) repair, and showed advantage to the latter approach in terms of morbidity and short term overall survival [12]. Nonetheless, EV repair might result in a higher rate of infection or recurrent bleeding [13,14,15]. Thus, the decision should be tailored to the specific patient, considering the timing of presentation and hemodynamic status, as well as the prognosis and status of underlying malignancy.

In conclusion, aorto-enteric fistula secondary to abdominal malignancy is a highly challenging medical condition. It is imperative to recognize patients at risk for this complication and to consider preventive intervention or treatment adjustment. Most importantly, its management requires a high index of suspicion and a skilled multidisciplinary team, including an oncologist, a blood bank specialist, a vascular surgeon, and an interventional radiologist, in order to increase the chance of a successful outcome.

The retrospective data collection was approved by the local ethics committee in accordance with Helsinki declaration standards.

## Figures and Tables

**Figure 1 diagnostics-11-00389-f001:**
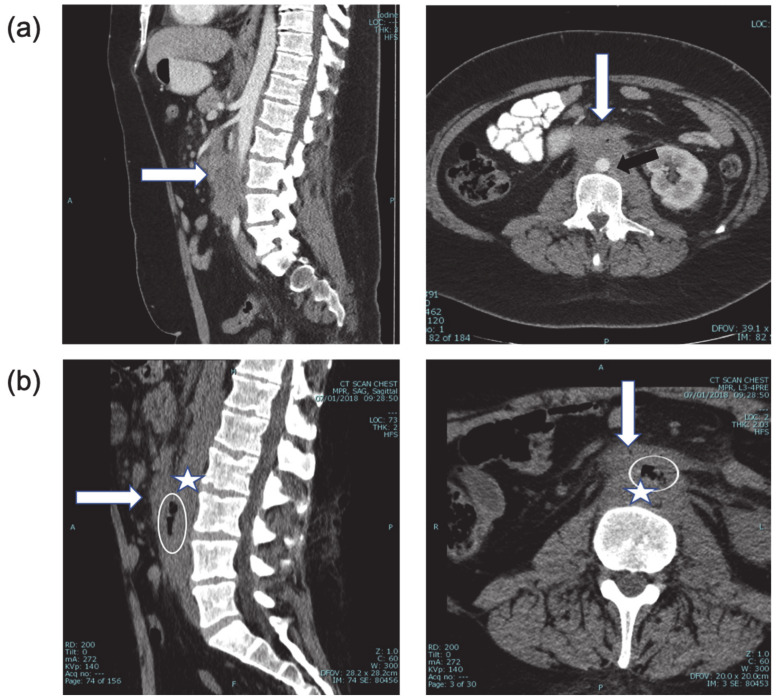
(**a**) CT + intravenous contrast, three months prior to bleeding event, showing the extent of disease (white arrow) around the abdominal aorta (black arrow) and relationship to duodenum (sagittal and axial views). (**b**) CT of the spine one month prior to bleeding. Sagittal and axial views. With retrospect, air (encircled) was identified between the aorta (star) and duodenum (arrow), highly suspicious for aorto-duodenal fistula.

**Figure 2 diagnostics-11-00389-f002:**
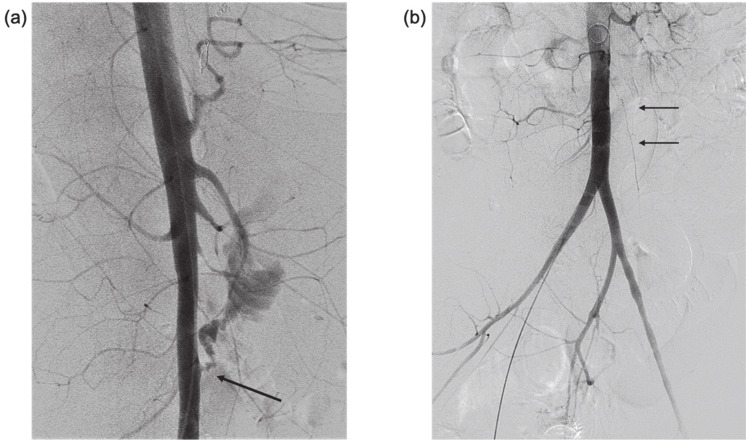
(**a**) Lateral digital subtraction aortography showing active contrast extravasation from the anterior infra renal aorta into small bowel lumen (arrow)—AEF. Note the diffuse small caliber branch vessels due to vasoconstriction in the setting of hemorrhagic shock. (**b**) Anterolateral aortogram after covered stent deployment in the infra renal aorta, covering the AEF (arrows). Note that extravasation is no longer seen, and there is some increase in vessel caliber due to improved hemodynamic status.

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
