# Peer review of "Diagnostic and Management Difficulty of Bleeding Aorto-Duodenal Fistula Associated with Hodgkin’s Lymphoma"

_diagnostics, 2021, doi:10.3390/diagnostics11030389_

Round 1

Reviewer 1 Report

The authors present an interesting case report of an aortoenteric fistula causing major GI bleeding in a patient with relapsed Hodgkin lymphoma. They highlight the diagnostic and management challenges with this entity. 

A few comments to address:

  1. Imaging details of relapsed Hodgkins lymphoma by PET -CT can be incorporated for clarity on extent of intra-abdominal adenopathy and proximity to the aorta. 
  2. How was the patient managed after the GI bleed for Hodgkins lymphoma - details of further therapies given?
  3. Why was a second autologous transplant done in this disease state? Was the patient considered for an allogeneic transplant?

Author Response

We thank the reviewer for the constructive comments

  1. Imaging details of relapsed Hodgkins lymphoma by PET -CT can be incorporated for clarity on extent of intra-abdominal adenopathy and proximity to the aorta. 

We have added CT images 3 months and 1 month prior to the bleeding event showing the extent of the disease and its proximity to the abdominal aorta, as well as findings suspicious for AEF prior to the event.

Line 43: we added a note to Figure 1. Figure 1 was changed to figure 2

Line 63: Figure 1. (a) CT + intravenous contrast, three months prior to bleeding event, showing the extent of disease (white arrow) around the abdominal aorta (black arrow) and relationship to duodenum (sagittal and axial views) (b) CT of the spine one month prior to bleeding. Sagittal and axial views. With retrospect, air [encircled] was identified between the aorta [star] and duodenum [arrow], highly suspicious for aorto-duodenal fistula.

2. How was the patient managed after the GI bleed for Hodgkins lymphoma - details of further therapies given?

3. Why was a second autologous transplant done in this disease state? Was the patient considered for an allogeneic transplant?

The following paragraph has been added to answer comment 2 and 3:

Line 86: The patient then received Nivolumab with partial response and proceeded to a second autologous bone marrow transplant with a cyclophosphamide/total body irradiation (Cy/TBI) conditioning. The decision to proceed to a second autologous transplant was made due to inadequate conditioning with melphalan in the first transplant, the rather low success rate with allogeneic transplant in Hodgkin’s lymphoma with active disease and the general condition of the patient. Imaging following the transplant there was an almost complete response at all of the sites of active disease, with residual disease in the lung.

Reviewer 2 Report

This is an interesting case report but I think the paper could benefit from a thorough review ot the literature. If the authors did not identify other cases of AEF due to Hodgkin's lymphoma, maybe they could also search for cases related to Non-Hodgkin's lymphoma. Maybe try searching in Google Scholar, Scopus, ScienceDirect rather than just PubMed-MEDLINE?

I think it's best to write Hodgkin's lymphoma rather than Hodgkin lymphoma. 

The cited manuscripts should be in brackets [...].

Revise the references to match the style of the journal, the DOI is missing. 

Author Response

We thank the reviewer for his constructive comments:

We have performed an extensive search for similar presentation in patients with lymphoma, including non-Hodgkin’s lymphoma and identified two more case reports with similar presentations, which were added to the introduction:

Line 26: Reports of AEF as a complication of lymphoma are rare.

Line 30: Two other reports linking described a patient with non-Hodgkin lymphoma involving the small bowel with suspected AEF [5], and a patient with Anaplastic T cell lymphoma in the mediastinum, who develop aorto-esophageal fistula, following initiation of treatment [6].

Line 109: Reference 6 was also added to the discussion to demonstrate another case in which the AEF presented with bleeding following initiation of treatment.

- I think it's best to write Hodgkin's lymphoma rather than Hodgkin lymphoma. 

We have made the suggested correction

- The cited manuscripts should be in brackets [...]. Revise the references to match the style of the journal, the DOI is missing.

References have been revised accordingly

Round 2

Reviewer 2 Report

The authors have addressed my suggestions and the paper can be accepted for publication.